# Polarization of Femtosecond Laser for Titanium Alloy Nanopatterning Influences Osteoblastic Differentiation

**DOI:** 10.3390/nano12101619

**Published:** 2022-05-10

**Authors:** Mathieu Maalouf, Alain Abou Khalil, Yoan Di Maio, Steve Papa, Xxx Sedao, Elisa Dalix, Sylvie Peyroche, Alain Guignandon, Virginie Dumas

**Affiliations:** 1SAINBIOSE Laboratory INSERM U1509, Jean Monnet University, University of Lyon, F-42270 Saint Priest en Jarez, France; steve.papa@univ-st-etienne.fr (S.P.); elisa.dalix@univ-st-etienne.fr (E.D.); sylvie.peyroche@univ-st-etienne.fr (S.P.); alain.guignandon@univ-st-etienne.fr (A.G.); 2Hubert-Curien Laboratory, Jean Monnet University, University of Lyon, UMR 5516 CNRS, F-42000 Saint-Etienne, France; alain.abou.khalil@univ-st-etienne.fr (A.A.K.); xxx.sedao@univ-st-etienne.fr (X.S.); 3GIE Manutech-USD, F-42000 Saint-Etienne, France; yoan.di-maio@manutech-usd.fr; 4Laboratory of Tribology and Systems Dynamics, Ecole Nationale d’Ingénieurs de Saint Etienne, Ecole Centrale de Lyon, University of Lyon, UMR 5513 CNRS, F-42100 Saint-Etienne, France; virginie.dumas@enise.fr

**Keywords:** femtosecond laser processing, multiscale patterning, human mesenchymal stem cell, cell adhesion, radial LIPSS

## Abstract

Ultrashort pulse lasers have significant advantages over conventional continuous wave and long pulse lasers for the texturing of metallic surfaces, especially for nanoscale surface structure patterning. Furthermore, ultrafast laser beam polarization allows for the precise control of the spatial alignment of nanotextures imprinted on titanium-based implant surfaces. In this article, we report the biological effect of beam polarization on human mesenchymal stem cell differentiation. We created, on polished titanium-6aluminum-4vanadium (Ti-6Al-4V) plates, a laser-induced periodic surface structure (LIPSS) using linear or azimuthal polarization of infrared beams to generate linear or radial LIPSS, respectively. The main difference between the two surfaces was the microstructural anisotropy of the linear LIPSS and the isotropy of the radial LIPSS. At 7 d post seeding, cells on the radial LIPSS surface showed the highest extracellular fibronectin production. At 14 days, qRT-PCR showed on the same surface an increase in osteogenesis-related genes, such as alkaline phosphatase and osterix. At 21 d, mineralization clusters indicative of final osteoinduction were more abundant on the radial LIPSS. Taken together, we identified that creating more isotropic than linear surfaces enhances cell differentiation, resulting in an improved osseointegration. Thus, the fine tuning of ultrashort pulse lasers may be a promising new route for the functionalization of medical implants.

## 1. Introduction

Titanium and related alloys, such as titanium-6aluminum-4vanadium (Ti-6Al-4V), have been used as the main biomaterial for dental and orthopedic implant devices, which can be attributed to their resistance to corrosion, mechanical strength and high biocompatibility with host tissues [1,2]. However, patients still face the risk of implant failure because of insufficient bone integration due to fibrous tissue production and/or the occurrence of infection [3]. It remains a challenge to improve osseointegration with good mineralization at the bone–implant interface. The initial cellular response to the implant surface is a key factor for successful and stable long-term integration [4]. Human mesenchymal stem cells (hMSCs) are attracted to the implant site and start interacting with the implant surface. This step is followed by cell proliferation and differentiation toward an osteoblastic lineage. Mature osteoblasts deposit an extracellular matrix, rich in collagen type I and fibronectin, and then produce a new calcified and mineralized matrix in contact with the implant surface [5]. The filled bone–implant interface of successfully integrated implants have a similar composition to natural mineralized bone.

Several attempts to optimize the osseointegration capacity of implant materials have been reported [6]. In particular, it has been shown to be possible to modify surface biofunctionalities, such as accelerated osteogenic differentiation, through surface roughening [7,8]. Surface roughness (Sa) in the microscale range (Sa = 1–2 μm) has been shown to increase bone generation with implant contact and has been described to improve cell osteogenic differentiation compared to smooth surfaces [9,10]. However, a surface with high microroughness can also enable bacterial infiltration, with the development of biofilm on the surface of the implant, leading to infection [11,12]. On the other hand, at smaller scales, nanostructures can help limit bacterial adhesion to the surface [13,14]. Hence, even more attention should be paid not only to microscale modifications, but also to the nanoscale patterning of implants (Sa < 100 nm) for accelerated osseointegration and limited biofilm. Nanostructuring of titanium surfaces via laser texturing is a very attractive and expanding area that should be explored further in great detail, as it has emerged as a powerful and versatile surface engineering process without additional coatings [15,16,17,18,19,20]. 

The femtosecond laser (FSL) offers the possibility to texture titanium at nanometric scales with controlled nanopatterns [15,21]. Particularly, periodic nanostructures such as laser-induced periodic surface structures (LIPSS) are formed during a complex interplay between incoming laser light and surface waves [22]. The periodicity of the LIPSS is closely linked to laser wavelength [23], and the direction of the LIPSS is determined by laser polarization [24]. As a consequence, LIPSS with desired periodicity and linear or radial organization can be produced by applying FSL light with a well-defined laser wavelength and polarization [21]. In the process of developing next-generation implants, it is of great importance to evaluate biological responses triggered by FSL-induced nanopatterns, which are dependent on surface parameters, including surface isotropy modification. 

Here, our goal was to use two different laser polarization states, linear or azimuthal, to create two differently designed patterns with nanometric dimensions on a Ti-6Al-4V surface. It is of interest to evaluate the influence of laser polarization on engineered surface patterns. The two nanopatterns were conceived in order to obtain a similar LIPSS density and similar nanoroughness; only the isotropy of the surface was modified. A linear laser beam polarization created an anisotropic surface (linear LIPSS), whereas an azimuthal polarization produced an isotropic surface (radial LIPSS). The idea of azimuthal and/or radial polarization for anisotropy breaking has been investigated previously [24], but the structures produced were in the µm scale. As mentioned earlier, µm-scale topography may favor bacterial infiltration; therefore, we aimed at isotropic patterning on a nanoscale. Furthermore, a direct comparison of isotropic nanostructures and anisotropic structures with identical roughness has not been studied thus far.

Human MSCs are multipotential bone-marrow-derived stem cells that can differentiate into a wide diversity of cell types, such as osteoblasts, and they represent a model of choice in the field of cell–material interaction [25]. It is known that hMSC osteoblastic differentiation can be directed by the complex interrelationship among surface properties of the material and the state of intracellular tension [25,26,27,28,29]. The isotropy of nanoroughness can be an important parameter to drive cellular responses. In order to engineer titanium surfaces with optimal properties for improved osteogenesis, it is essential to understand if cellular responses are modified by isotropic or anisotropic surface nanoroughness.

In this in vitro study, we assessed several stages of the osseointegration process. By providing aligned or disordered textures to hMSCs, we hypothesized that cells differentially organize their focal adhesions and, more importantly, their fibrillar adhesion (increased cell contractility), thus accelerating their differentiation toward osteoblasts [30]. Moreover, alterations in extracellular matrix mechanosensing are known to be potent regulators of osteogenesis, bone formation and implant maintenance. RGD-containing ligands (such as fibronectin and osteopontin) present in remnant bones support the adhesion, spreading and conversion of marrow-derived stem cells into osteoblasts [31,32]. These processes are associated with the overexpression of several genes involved in osteogenesis, such as osterix, bone sialoprotein and alkaline phosphatase. Finally, the ultimate step is initiation and mineralization. Based on the importance of all these different events, we first investigated the influence of linear or radial LIPSS on cell adhesion and contractility 24 h post seeding by assessing focal adhesion characteristics. Then, fibronectin production was quantified at day 7, followed by the expression of osteogenic-related genes at day 14 and mineralization surfaces at day 21, compared to conventional polished titanium alloys. Our results clearly indicate that isotropic texturing improved the induction of osteogenesis on titanium surfaces at all investigated time points. 

## 2. Materials and Methods

### 2.1. Titanium Alloy Samples

Mirror-polished titanium alloy samples of Ti6Al4V from Goodfellow (Huntingdon, UK) were used in these experiments. The samples exhibited square dimensions of 1 cm^2^ and a thickness of 1 mm with a roughness of Ra = 0.5 µm.

### 2.2. Laser Surface Texturing

Titanium samples were textured using FSLs from Amplitude Systems and galvo scanners from Scanlab within the GIE Manutech-USD platform (Saint-Etienne, France). The following laser sources were used: a Tangor HP and a Tangerine FSL (both from Amplitude Laser Groupe, Pessac, France), operating at 1030 nm central wavelength and a pulse duration of around 400 fs. Both lasers exhibited linear polarization states. All samples were placed on XYZ translation stages from Aerotech in order to find the best focusing plane. Scanlab GmbH’s intelliSCAN 14 scanners were finally associated with two different f-theta lenses of 56 mm and 100 mm depending on the targeted texturing. 

Different types of structures were realized using such lasers (Table 1):IR linear LIPSS: They were generated on titanium alloy surfaces using the Tangor and Tangerine lasers and the 100-mm f-theta lens with a fluence of between 0.3–0.47 J/cm^2^. The fluence peak is defined as: $F_{peak}=\frac{2\;E}{\pi \;\omega_{0}{}^{2}}$. Spacing between pulses and a hatch distance of 4–5 µm were chosen with a 16-µm measured 1/e² beam diameter. This configuration makes a number of effective pulses per spot diameter of Neff_1D  =  4 and a line separation distance of Δ = 4–5 µm.IR radial LIPSS: A 56-mm f-theta lens was deployed, leading to a 11-µm measured 1/e² beam diameter. In order to create an exotic LIPSS with ripples pointing to the outside of the center of the beam, an s-wave plate was implemented in the beam path of the laser to create a donut-shaped beam that converted the entering linear polarization into an azimuthal polarization. This decision was based on previous experiments on the behavior of cell adhesion with this type of structure. A laser pulse energy *E* of 0.7 µJ was applied to obtain the radial LIPSS. Impact positions were defined in order to keep the isotropic effect provided by such nanostructures. As a consequence, a 13-µm spacing between pulses led to tangency with a slight overlay on the edges of the impacts. An accumulation of 5 pulses per impact was chosen to compensate for a lack of energy accumulation due to this low recovery rate. It is worth mentioning that the work field of this f-theta lens is about 50 × 50 mm^2^ in area. In the current study, we produced our radially aligned LIPSS by using azimuthal polarization in a relatively small area of 10 × 10 mm^2^. No marked influence of mirror positions upon LIPSS quality was wittenessed in such a case. The same nominated laser conditions proceduced similar impacts at the extremity.

### 2.3. Surface Morphology

Knowing that the scale of the nanostructures created by the laser is near the resolution limit of an optical microscope (~550 nm), scanning electron microscopy (SEM) was used to visualize and characterize the different laser-induced patterns. A Tescan VEGA3 SB, Brno Czech Republic electron microscope was used, operating at 20 kV and with a secondary electron detector. 

### 2.4. Surface Topography

Atomic force microscopy (AFM, JPK Instruments, Berlin, Germany) was used to characterize the different nanostructures. Data topographies were analyzed with Mountains Map^®^ 8.2 software. Surface roughness parameters were computed on the treated surfaces, and this study focused on the following:Areal arithmetic mean height Sa (nm), which expresses the difference in height of each point compared to the arithmetical mean of the surface. This parameter is used to evaluate surface roughness.Texture aspect ratio Str, which expresses the isotropy and anisotropy of the topography. Str is a value ranging from 0 to 1. A value close to 0 indicates directionality (anisotropy), whereas a value close to 1 indicates that the surface does not exhibit preferred directions (isotropy).Polar spectrum, which shows the privileged texture directions.Average period and depth of the ripples.Ripple density per cm^2^.

The definitions of the surface parameters are stated in ISO 25178 standards [33].

### 2.5. Cell Culture

Prior to the cell culture, we applied the sterilization procedure of industrial standard to the laser-irradiated samples. In summary, titanium samples were autoclaved at 134 °C for 19 min. Human MSCs from PromoCell (hMSC-BM-c, C-12974) at passage 4 were maintained in a T75-flask for 3 d in a growth medium (MSCGM, C-28009, PromoCell). Cells were then seeded on titanium samples at 7000 cells/cm^2^ in 24-well plates with a growth medium. At 24 h post seeding, the growth medium was replaced with an osteogenic medium (MSCODM, C-28013, PromoCell). Thereafter, the osteogenic medium was renewed every 4 d.

### 2.6. Fluorescent Cell Labeling at 24 h and 7 d Post Seeding

At 24 h post seeding, some samples of each type were fixed in 10% formalin for 30 min at room temperature and then permeabilized with 0.1% Triton X-100 in phosphate-buffered saline (PBS) for 3 min. Samples were incubated with rhodamine-conjugated phalloidin diluted at 1:300 in PBS at 37 °C for 1.5 h for actin labeling (cytoskeleton). Then, focal adhesion labeling was performed using a fluorescein isothiocyanate (FITC)-conjugated vinculin antibody (Prod. No. F7053, Sigma-Aldrich, St. Louis, MO, USA) diluted at 1:50 in PBS at 4 °C overnight. Afterwards, nuclei labeling was performed with 1 µg/mL 40, 6-diamidino-2-phenylindole (DAPI) diluted at 1:200 in PBS at room temperature for 20 min. Washes were performed using PBS between each step of the experiment.

At 7 d post seeding, other samples of each type were fixed and permeabilized in 10% formalin and 0.1% Triton X-100, respectively, as described above. Samples were incubated with rhodamine-conjugated phalloidin diluted at 1:300 in PBS at 37 °C for 1.5 h for actin labeling. Cells were then incubated with fibronectin (extracellular glycoprotein for cell adhesion) antibody, diluted at 1:100 in PBS at 37 °C for 2 h. Then, the samples were incubated with 488 fluoprobes diluted at 1:250 in PBS for 1.5 h at room temperature. Finally, nuclei labeling was performed with DAPI, as explained above. Washes were performed using PBS between each step.

### 2.7. Quantitative Real Time PCR (qRT-qPCR) at 14 d Post Seeding

For qRT-PCR, the cells were harvested on the laser-patterned and reference-polished titanium surfaces with Tri-Reagent (Sigma-Aldrich). RNA amounts were assessed with the Ribogreen kit (Invitrogen, Life Technologies, Eugene, OR, USA), and their quality checked with the Experion-automated electrophoresis station (Bio Rad, Hercules, CA, USA). Messenger RNA was reverse transcribed (iScript cDNA synthesis Kit, Biorad) according to the manufacturer’s instructions, and then 300 ng of cDNA was amplified through qRT-PCR using the SYBR Green I dye (Lightcycler faststart DNA masterSYBR green I, Roche, Germany). Primer sequences for the osteogenic genes of interest are given in Table 2. The expression of the housekeeping gene (GAPDH) did not vary significantly within or between groups in either experimental setting (data not shown).

### 2.8. Assessment of Mineralization at 21 d Post Seeding

At 21 d post seeding, the cells on the different surfaces were fixed with formalin for 30 min and then were washed with demineralized water. A 3 × 10^−3^ M solution of calcein blue (M1255-10G, Sigma) was deposited on the cells and left for 5 min at room temperature. The samples were then rinsed with demineralized water and dried. 

### 2.9. Image Acquisition and Analysis

At 24 h, 7 d and 21 d, 8 fields of 0.7 mm^2^ were acquired for each surface using a confocal laser microscope (Zeiss LSM 800 Airyscan, Oberkochen, Germany) equipped with Zen software. DAPI labeling (at 24 h and 7 d) and calcein blue labeling (at 21 d) were visualized with a 405-nm-wavelength laser. Vinculin (at 24 h) and fibronectin (at 7 d) labelings were visualized with a 488-nm laser. Actin labeling (at 24 h and 7 d) was visualized with a 561-nm laser. All images were analyzed with imageJ software to obtain cell density and contractility (cell area/vinculin area) at 24 h post seeding, fibronectin area at 7 d, and relative mineralized area as well as the number of mineralized spots and average mineralized area at 21 d.

### 2.10. Statistics

The data were compared between each surface by performing a Wilcoxon–Mann–Whitney *U*-test. Significance was set at *p* = 0.05. The comparison between the polished surface and the two textured ones aimed at evaluating the effect of each texture compared to a standard reference. The comparison between linear LIPSS and radial LIPSS was performed to assess the effect of the laser beam polarization change.

## 3. Results

### 3.1. Directionality of LIPSS Was the Main Difference between the Two Designed Textures 

The two surface structures manufactured in the present work were LIPSS, produced by a radiation wavelength of λ 1. SEM images in Figure 1A showed that linear laser polarization produced linear periodic LIPSS, whereas azimuthal polarization produced LIPSS with a radial direction. The directions of the structures of LIPSS were perpendicular to the polarization direction of the laser beam.

AFM measurements enabled a 3D-view reconstruction of the surface topography and the calculation of different surface parameters with Mountains Map^®^ software (Figure 1B). The linear LIPSS texture has a surface roughness equivalent to the radial LIPSS texture, with Sa of 71 nm and 77 nm, respectivelly. The average roughness was, however, not sufficient to uniquely describe a surface in relation to cellular behavior. The depth of ripples (151 nm vs. 130 nm) and density of ripples (22309/cm² vs. 21381/cm²) were also very similar between the two laser-textured structure types. However, large differences in texture direction could be observed. The texture aspect ratio Str was close to 0 for the linear LIPSS (0.083), indicating an anisotropic surface with directionality. This was confirmed by the polar spectrum, which showed a privileged direction of 90°. In contrast, the radial LIPSS had a Str value close to 1 (0.682), underlining the isotropy of the surface, and the polar spectrum did not indicate a privileged direction. All these data on surface parameters indicated that the main difference between the linear LIPSS and radial LIPSS lay in the directionality of the overall texture.

### 3.2. Radial LIPSS Increased Cell Contractility in the Early Stages

Immunostaining performed 24 h post seeding showed that cell density was similar among the three surfaces, indicating that hMSCs have the same probability of attachment on all three textures (Figure 2). Vinculin, a major protein present at focal adhesion complexes, showed a different pattern between the three surfaces. The cells on polished and linear LIPSS surfaces presented a higher number of focal adhesions (indicated by the white arrows and the zoom squares in Figure 2) compared to the radial LIPSS surface. In addition, the polished surface showed the most prominent focal adhesions. As we know that vinculin contacts may undergo rapid turnover, we compared the measurements of mean cell area/mean vinculin area ratio as an indirect evaluation of focal adhesion dynamic or turnover. The ability of a cell to maintain its shape with small contacts relies on increased cell tension or contractility. We found significantly higher cell contractility on the radial LIPSS surface compared to the two other surfaces (+467% vs. polished; +391% vs. linear LIPSS; *p* = 0.00078 for both). 

### 3.3. Radial LIPSS Improved Fibronectin Matrix Production

At 7 d post seeding, fibronectin production was assessed by immunolabeling to observe the advancement of extracellular matrix production. Fibronectin is a protein secreted in the extracellular matrix that serves as an attachment of the cells to several components, such as collagen fibers. Moreover, fibronectin is required for osteoblast mineralization. Image analysis at 7 d showed that the radial LIPSS surface displayed a higher amount of extracellular fibronectin proteins compared to polished (+16%, *p* = 0.045) and linear LIPSS (+31%, *p* = 0.016) surfaces (Figure 3). 

### 3.4. Radial LIPSS Induced Overexpression of Osteogenic Related Genes

At 14 d post seeding, qRT-PCR analysis showed that the radial LIPSS surface induced an increase in the osteoblastic differentiation genes OSX (+465% vs. polished, *p* = 0.009 and +252% vs. linear LIPSS, *p* = 0.045) and BSP (+91% vs. polished, *p* = 0.047) (Figure 4). In addition, matrix production genes were also increased on the radial LIPSS surface, such as fibronectin (+23% vs. polished, *p* = 0.027), which is in line with the observations made at 7 d, and COL1A1 (increased in trend compared to the polished surface). Furthermore, mineralization genes such as ALP and OPN were increased on the radial LIPSS surface compared to the polished surface (Figure 4). OCN, a major gene for mineralization processes, was not different among the three surfaces, potentially related to the early timing of qRT-PCR assessment (14 d) in relation to the expression of this gene. There was no increase in osteogenic genes for the linear LIPSS surface compared to the polished surface.

### 3.5. Radial LIPSS Increased the Mineralized Surfaces

At 21 d post seeding, mineralization was assessed by calcein blue labeling. Image analysis revealed that hMSCs on the radial LIPSS surface had increased mineralization activity compared to the other two surfaces (Figure 5). This was highlighted by a larger mineralized area (+99% vs. polished, *p* = 0.016; +132% vs. linear LIPSS, *p* = 0.002) and a higher number of mineralized spots (+83% vs. polished, *p* = 0.046; +128% vs. linear LIPSS, *p* = 0.006) on the radial LIPSS surface. The average size of the mineralization spots was not different between the three surfaces.

## 4. Discussion

The topography of titanium is one of the key features for the acceleration of osteogenic cell differentiation on medical implant devices. Stem cells interact with underlying surface patterns, which lead to modulating the cell’s fate [34,35]. In this study, we focused our work on two different nanostructures obtained by FSL texturing. The nanoroughness of the two textures was similar, but one texture was anisotropic (linear LIPSS), whereas the other could be considered as isotropic (radial LIPSS). We have demonstrated that isotropic texturing improves osteoblastic differentiation compared to anisotropic and polished surfaces. This was conducted by a cross-time investigation and by various operational techniques. 

Our data demonstrate that a concerted assembly of fibronectin networks and a commitment toward osteoblastogenesis occurs earlier on irregular nanopatterned surfaces (isotropic) as compared to regular or untextured ones. As nicely demonstrated by Han et al., polished and anisotropic surfaces lead to more abundant focal adhesion with longer lifetimes, suggestive of higher recruitment and activation of FAK and Src as a response to increased integrin binding and clustering, and with enhanced osteogenesis [30]. We were surprised to find that initial smaller contacts (vinculin positive) were formed on the most osteoinductive surface. Our data showing significantly higher accumulations of fibronectin on the texture presenting the smallest contacts are somewhat expected, given that smaller vinculin contacts are not associated with cell adhesion defects or reduced contractility. On the other hand, cells on isotropic surface (radial LIPSS) presented numerous actin fibers as well as increased ratios of cell area/vinculin area and thus increased contractility. We can speculate that maintaining efficient cell spreading on isotropic surfaces necessitates highly dynamic focal contacts (increased turnover), leading to higher maturation in fibrillar contacts, potentially explaining increased fibronectin fibrillogenesis [36]. This may explain sustained integrin signaling; enhanced mechanotransduction and osteogenesis, as evidenced by gene expression levels of OSX, OPN and ALP; and the deposition of mineral clusters. It is possible that on isotropic surfaces, cells adapt to increased cell contractility by accumulation and force transmission into the actin cytoskeleton, leading to increases in YAP/TAZ nuclear accumulation, which coactivates with RUNX2 (and TEAD) and leads to improved osteogenic commitment [37].

In many studies, one of the most investigated aspects related to topography is surface roughness (Ra (2D parameter) or Sa (3D parameter)), which quantifies the protrusions or depressions on the surface [38]. Sa can be considered an extension of Ra (arithmetical mean height of a line). It expresses, as an absolute value, the difference in height of each point compared to the arithmetical mean of the surface. Concerning implant surfaces, roughness can be divided into different levels: macroroughness (Ra scale around 10 μm), microroughness (Ra scale around 1 μm) and nanoroughness (Ra scale < 200 nm). In general, macroroughness is described to improve bone–implant interactions, but it is well known that the optimal surface roughness should be about 1 µm to promote the osteogenic differentiation of MSCs [39,40]. Unfortunately, increasing surface roughness can promote the microbial colonization of the implant surface, as initial bacterial adhesion begins at sites that offer shelter [41,42]. Several studies have shown that, in order to reduce bacterial adhesion, a nanoscale surface with a roughness of <200 nm is the most effective, since the depth and period of the nanostructures are smaller than most bacteria [12,42,43]. Moreover, a nanoscale surface roughness can also influence stem cell fate. Although the optimal scale of nanoroughness for osteogenic differentiation is challenging to define, it has generally been shown that nanoroughness can also enhance osteogenic differentiation [44,45,46]. Previous studies have reported osteogenic performance depending on nanofeature morphologies or organization, such as anisotropic or isotropic nanotextures [47].

In our study, in order to link osteogenic differentiation to surface nanoroughness, we used a classification based on the orientation of topography (isotropic or anisotropic). An anisotropic surface is a surface with a specific orientation, such as a linear LIPSS. In contrast, an isotropic surface is a surface with no preferred orientation, such as a radial LIPSS. Anisotropic surfaces have often been studied in terms of a tool to direct cell alignment, which can influence stem cell fate. On substrate surfaces with groove scales < 500 nm, studies have reported that MSCs are committed to adipogenic and myogenic lineages [48,49]. On the other hand, osteogenic differentiation can be decreased by anisotropic textures, since as microscale periodicity becomes lower, osteogenesis also lowers [50]. An isotropic surface is not described as influencing cell alignments, but it is proved to control cell functions. Isotropic topographies such as nanopillar or nano-island usually enhance osteogenic differentiation, but there is a lack of clear data to establish which precise size or density of nanopattern is more efficient [51,52,53]. Cell osteogenic responses to isotropic patterns are often inconsistent because dimensions, organization or density of nanopatterns vary from one study to another. Only one study from Dalby et al. verified how the distribution of topographical features influences cell differentiation [39]. They reported that surfaces composed of nanopits with controlled disorder resulted in increased osteogenic markers compared to highly ordered surfaces or randomly displaced nanopits. 

Very strict surface chemistry analysis and cytotoxicity tests are to be made. Nonetheless, as these are considerably time consuming, they are to be pursued and performed on surface textures that already exhibit a promising osteogenesis property after bioassessment. If necessary, anodization of our laser-treated samples can be considered for shielding out surface chemistry changes, if any, induced by laser irradiation. These topics deserve separate and dedicated studies. They are, naturally, beyond the scope of this report, but they are not to be left uninvestigated in future developments. We are well aware that the chemistry of extreme surfaces is inevitably altered after laser irradiation in air, such as observations made in other studies [24,54,55,56]. Even though our irradiation condition applied in this study is even milder than the mildest one mentioned in the literature, there is no means of knowing if we are not producing surface chemistry modification to some extent.

Finally, in terms of surface nanotexturing, the most common techniques on titanium, such as acid etching, cluster deposition, sandblasting and anodizing, yield to the limited control and adjustability of surface features [57,58,59,60]. Thus, it can be difficult to obtain repeatable cell behavior due to batch-to-batch fluctuations. More accurate monitoring of nanopatterns typically requires lithographic techniques, but these are challenging to apply on materials such as titanium [51,61]. Otherwise, FSLs allow for the creation of reproducible features. With a precise tuning of laser parameters, such surfaces can be used to produce well-controlled cell effects, paving the way to unravel the complex interplay between cells and topography. Ultrashort pulsed laser texturing has emerged as a powerful and versatile surface engineering process. It is a highly valid and innovative approach and is more eco-friendly due to its simplicity and its flexibility, with no consumables except light itself. Such a solution can also be performed in air environments, which further makes it a good candidate for implementation in an industrial context for functionalizing orthopedic and/or dental implants. As mentioned before, the possible variation of surface chemistry after laser irradiation is to be investigated, and/or surface annodization (such as the method described in [62]) for shielding away any possible surface chemistry changes is to be performed.

## 5. Conclusions

This study provides insights into how small changes in laser polarization can modulate hMSC fate choice and extracellular matrix deposition. Nanoscale radial LIPSS generated on titanium alloys by FSLs with azimuthal polarizations exhibited an isotropic distribution that seemed to promote osteoblastic differentiation. Cell contractility and density, fibronectin production, gene overexpression and improved mineralization helped to ensure the consistency of cell evolution analysis over a time period of several weeks. Such dynamic hMSC sensitivity appears very important during implant osteogenesis, as these cells are primarily recruited to the implantation site. As a perspective, texturing titanium surfaces with such small features opens the possibility to couple-enhanced osseointegration with potential antibacterial properties, which are known to be more sensitive at the nanoscale than the microscale.

## Figures and Tables

**Figure 1 nanomaterials-12-01619-f001:**
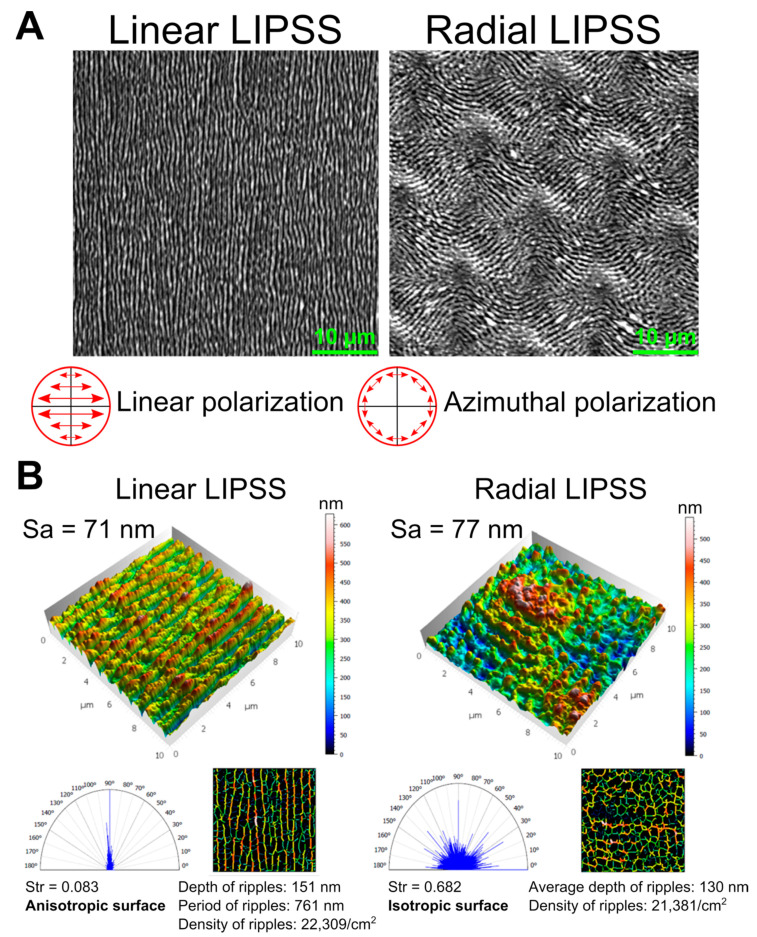
(**A**) SEM images of the two surface textures with linear or radial LIPSS (magnification 3000×). (**B**) 3D image reconstruction of the nanometer-scale surface texture with linear or radial LIPSS (data from atomic force microscopy images 10 µm × 10 µm) and surface analysis by Mountain Map^®^ software.

**Figure 2 nanomaterials-12-01619-f002:**
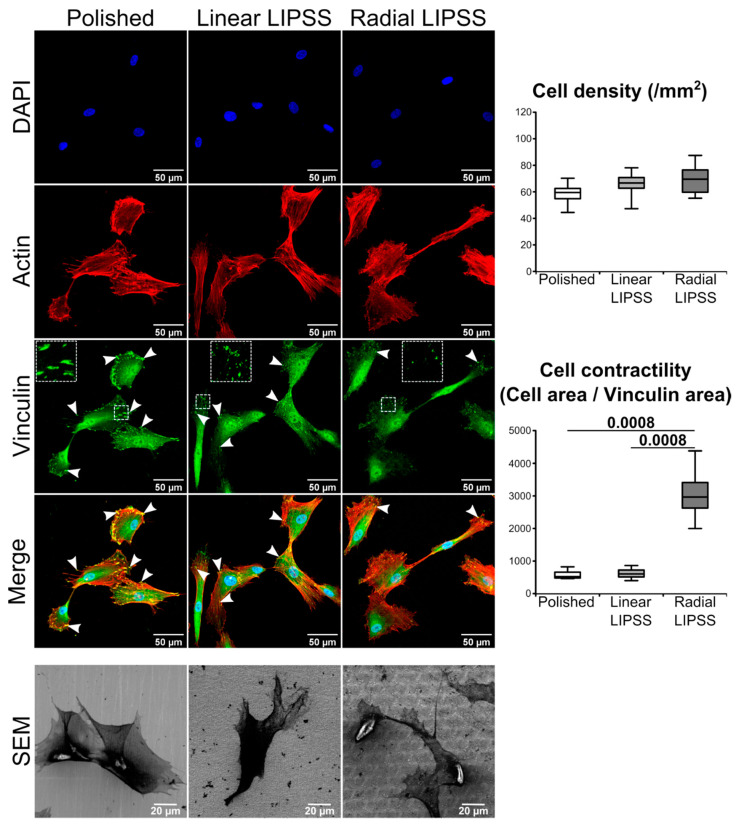
DAPI nuclear staining (blue), rhodamine-conjugated phalloidin-labeled actin (red), FITC-conjugated vinculin antibody (green), overlaid fluorescent image of immunostained cellular component (merged) for the cells cultured on polished, linear LIPSS and radial LIPSS surfaces 24 h post seeding. SEM images show other cells present on the three surfaces. White arrows indicate focal adhesions recognized by prominent vinculin labeling. The squares show a zoom of some focal adhesions for each surface. On the right side of the image are the results of cell density and contractility for the three surfaces. The graphs show the *p*-values between each group, Mann–Whitney *U* test, *n* = 8 fields/group.

**Figure 3 nanomaterials-12-01619-f003:**
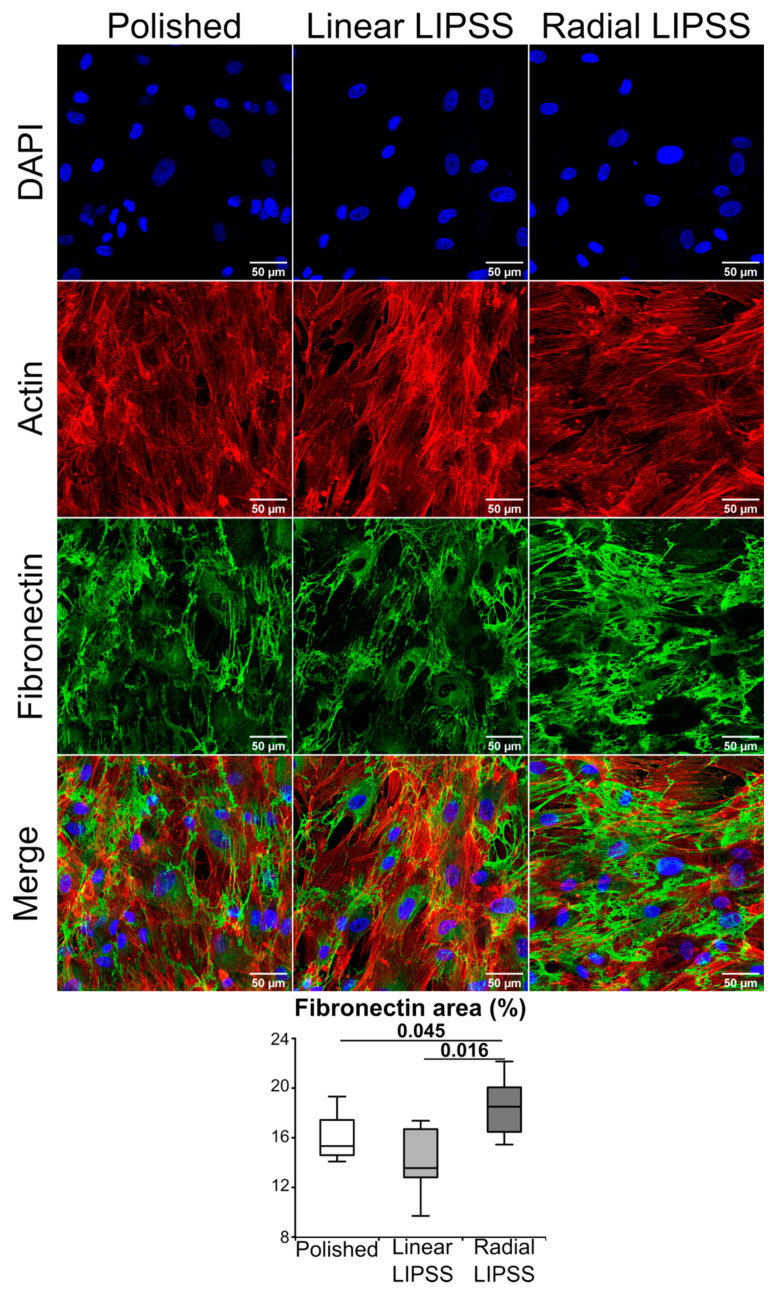
DAPI nuclear staining (blue), rhodamine-conjugated phalloidin labeled actin (red), fibronectin antibody coupled with a 488 fluoroprobe (green) and overlaid fluorescent image of immunostained cellular component (merged) for the cells cultured on polished linear LIPSS and radial LIPSS surfaces 7 d post seeding. On the bottom of the image is the result of the fibronectin area for the three surfaces. The graph gives the *p*-values between each group, Mann–Whitney *U* test, *n* = 8 fields/group.

**Figure 4 nanomaterials-12-01619-f004:**
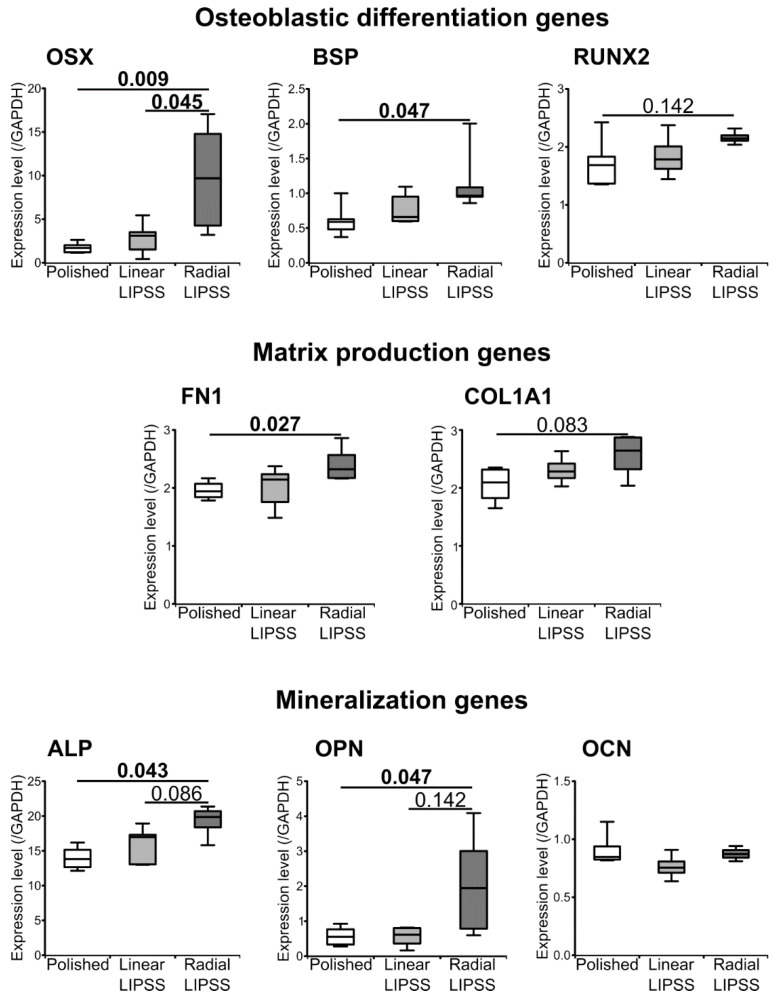
Osteogenic-related genes analyzed by RT-qPCR for cells present on polished, linear LIPSS and radial LIPSS surfaces at 14 d post seeding. Genes are classified by biological function. Values are given as ratios to GAPDH. The graphs give *p*-values between each group, in **bold** when <0.05, Mann–Whitney *U* test, *n* = 4–6 samples/group.

**Figure 5 nanomaterials-12-01619-f005:**
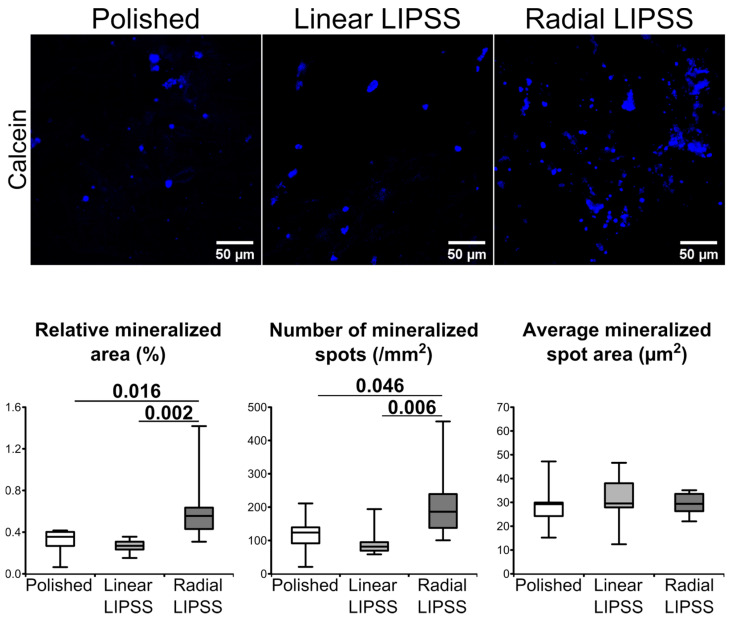
Mineralization clusters labeled by calcein blue (blue) on the polished linear LIPSS and radial LIPSS surfaces at 21 d post seeding. On the bottom of the image are the results of the different quantifications made for the three surfaces. The graphs show the *p*-values between each group, Mann–Whitney *U* test, *n* = 8 fields/group.

**Table 1 nanomaterials-12-01619-t001:** Laser parameters used to create the different titanium alloy textured surfaces.

Topographies	Laser Pulse Condition	Distance between Pulses	Hatching Distance	F-Theta
Linear LIPSS	$F_{\mathrm{peak}}$ = 0.3–0.47 J/cm^2^	4–5 µm	4–5 µm	100 mm
Radial LIPSS	E = 0.7 µJ/pulse 5 pulses/impact	13 µm	13 µm	56 mm

**Table 2 nanomaterials-12-01619-t002:** PCR primer sequences of genes implicated in different osteogenic pathways.

Protein	Forward	Reverse	Gene Bank ID
ALP	tgtaaggacatcgcctacca	gaagctcttccaggtgtcaa	NM_000478.5
BSP	gaagactctgaggctgagaa	cctctgtgctgttggtactg	NM_004967.3
COL1A1	tccggctcctgctcctctta	gttgtcgcagacgcagatcc	NM_000088
FN1	ggctggatgatggtagattg	tgcctctcacacttccactc	NM_212482.4
GAPDH	catcaccatcttccaggagcga	gtggtcatgagtccttccacga	NM_001289745.1
OCN	agcggtgcagagtccagcaa	agccgatgtggtcagccaac	NM_199173.5
OPN	tgatggccgaggtgatagtg	atcagaaggcgcgttcaggt	NM_001251830.1
OSX	ctggctgcggcaaggtgtat	ccagctcatccgaacgagtg	NM_001300837.1
RUNX2	ccttgaccataaccgtcttc	aaggacttggtgcagagttc	NM_001024630.3

Abbreviations: ALP: alkaline phosphatase; BSP: bone sialoprotein; COL1A1: collagen type 1 α1 chain; FN1: fibronectin 1; GAPDH: glyceraldehyde-3-phosphate dehydrogenase (housekeeping gene); OCN: osteocalcin; OPN: osteopontin; OSX: osterix; RUNX2: runt-related transcription factor 2.

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
