# Peer review of "Polarization of Femtosecond Laser for Titanium Alloy Nanopatterning Influences Osteoblastic Differentiation"

_nanomaterials, 2022, doi:10.3390/nano12101619_

Round 1

Reviewer 1 Report

The manuscript reported the biological effect of beam polarization on human mesenchymal stem cells differentiation. Laser Induced Periodic Surface Structure (LIPSS) was created on polished Titanium-6Aluminum-4Vanadium (Ti-6Al-4V) plates by using linear or azimuthal polarization of Infra-Red beams. The results show that creating more isotropic surfaces than linear enhances cell differentiation resulting in an improved osseointegration and ultrashort pulse lasers may be a promising novel way for unctionalization of medical implants. I’d recommend publishing the manuscript after addressing the following points:

  1. In introduction, the author claimed “The period and the direction of the nanostructures such as Laser-Induced Periodic Surface Structures (LIPSS) with linear or radial organization can be mastered thanks to the laser’s wavelength and polarization.” How authors can explain this influence of the laser’s wavelength and polarization to the nanostructures?
  2. On line 254, the sentence show “The cells on polished and linear LIPSS surfaces presented a higher number of focal adhesions (indicated by the white arrows and the zoom squares in Figure 2) compared to the radial LIPSS surface.” Why linear LIPSS surfaces presented a higher number of focal adhesions than the radial LIPSS surface?
  3. In figure 3, Image analysis at 7 days showed that the Radial LIPSS surface displayed a higher amount of extracellular fibronectin protein compared to polished (+16 %, p=0.045) and linear LIPSS (+31 %, p=0.016) surfaces. What reason caused a higher amount of extracellular fibronectin protein?
  4. The author claimed “Stem cells interact with underlying surface patterns 319 which lead to modulate the cell fate.” Please give detailed description or relevant literature for better understanding.
  5. Please check the full paper carefully to correct grammatical and spelling errors

Author Response

Dear reviewer,

Please find attached our responses and modifications to the article

Reviewer 2 Report

In this manuscript, the authors demonstrated a Linear Laser Induced Periodic Surface Structure (Linear LIPSS) and a Radial LIPSS developed by adjusting the polarization of ultrafast laser beam. Cell experiments confirmed isotropic surface can enhance cell differentiation and improve osseointegration. I suggest the publication of this manuscript after some minor revisions and answering following questions.

  1. The authors confirmed that isotropic structure can promote osteoblastic differentiation by experiment, but the theoretical analysis is too qualitative and simple.
  2. Line 261-262: “We found significantly higher cell contractility on the radial LIPSS surface compared to the two other surfaces.”. The underlying reason should be given in the main text of this manuscript.
  3. The size of text in all the Figures should be adjusted.
  4. There are some grammatical mistakes. The English needed to be polished.

Author Response

(The authors gave the same response as above.)

Reviewer 3 Report

The authors present a paper on the influence of different laser-induced periodic surface structures on medical grade titanium alloy to characterize the growth of cells (osteoblasts) on the laser treated surfaces and unirradiated polished areas. The irradiations were produced with linear and azimuthal polarization, producing regular low spatial frequency LIPSS and structures the authors call “radial” LIPSS. AFM and SEM characterizations were used for morphological characterization. Detailed cell studies are presented on the presence of vinculin and fibronectin of cells on the different areas with correlation to the cell adhesion. The results suggest that radial LIPSS allow the best conditions for cell adhesion and density. I find the paper is complete and the findings are of use for the scientific community on the field, however, there are some additional comments that should be included before the paper is considered for publication in Nanomaterials.

On the introduction, the authors missed citing numerous works on the subject regarding irradiations with femtosecond laser pulses with radial and azimuthal polarization, studies on Ti allow with femtosecond lasers and their chemical study, reports on the use of the Ti alloy for dental implants with periodic structures for the growth of osteoblasts, and the changes in the wetting properties that the polarization may induce when LIPSS use different than linear polarization, in detail the following papers:

Skoulas, E., Manousaki, A., Fotakis, C., & Stratakis, E. (2017). Biomimetic surface structuring using cylindrical vector femtosecond laser beams. Scientific Reports, 7(March), 45114. https://doi.org/10.1038/srep45114

Florian, C., Wonneberger, R., Undisz, A., Kirner, S. v., Wasmuth, K., Spaltmann, D., Krüger, J., & Bonse, J. (2020). Chemical effects during the formation of various types of femtosecond laser-generated surface structures on titanium alloy. Applied Physics A, 126(4), 266. https://doi.org/10.1007/s00339-020-3434-7

Kirner, S. V., Wirth, T., Sturm, H., Krüger, J., & Bonse, J. (2017). Nanometer-resolved chemical analyses of femtosecond laser-induced periodic surface structures on titanium. Journal of Applied Physics, 122(10), 104901. https://doi.org/10.1063/1.4993128

Zwahr, C., Welle, A., Weingärtner, T., Heinemann, C., Kruppke, B., Gulow, N., Holthaus, M. große, & Fabián Lasagni, A. (2019). Ultrashort Pulsed Laser Surface Patterning of Titanium to Improve Osseointegration of Dental Implants. Advanced Engineering Materials, 1900639, 1–11. https://doi.org/10.1002/adem.201900639

Heitz, J., Plamadeala, C., Muck, M., Armbruster, O., Baumgartner, W., Weth, A., Steinwender, C., Blessberger, H., Kellermair, J., Kirner, S. v., Krüger, J., Bonse, J., Guntner, A. S., & Hassel, A. W. (2017). Femtosecond laser-induced microstructures on Ti substrates for reduced cell adhesion. Applied Physics A: Materials Science and Processing, 123(12), 0. https://doi.org/10.1007/s00339-017-1352-0

Florian, C., Skoulas, E., Puerto, D., Mimidis, A., Stratakis, E., Solis, J., & Siegel, J. (2018). Controlling the Wettability of Steel Surfaces Processed with Femtosecond Laser Pulses. ACS Applied Materials & Interfaces, 10(42), 36564–36571. https://doi.org/10.1021/acsami.8b13908

Importantly, it has been recently demonstrated that oxidation prone materials do oxidate when irradiated under ambient conditions with femtosecond laser pulses (see Kirner, S 2017 and Florian, C. 2010, from the selection above). This oxidation impacts directly the chemistry at the surface affecting how different types of cells stick to a specific surface. Here, in the present study, the authors fail to name previous studies on the subject and assume the growth of cells is only driven by morphological changes omitting completely the chemical modifications induced by the irradiation. Considering the amount of available literature on the subject, it is a missed opportunity to develop further or at least comment its impact.

Additional comments are needed on the generation of azimuthal polarization. It is true that using an s-plate can produce it without further do, however, scanners from ScanLab normally use two crossed galvanometric mirrors that might affect how polarization is deployed towards the f-theta lens. These modifications can produce azimuthal polarization of ‘good quality’ only in a reduced area at the center of the irradiation area. Did the authors perform experiments that confirm the polarization over the areas produced for the subsequent characterizations?

Minor changes:

  • In line 71: FSL induced nano patterns.
  • The authors state that the ‘isotropic’ surface is covered by “radial” structures, which is not in line with the meaning of ‘isotropic’ where no clear direction differentiation can be made. I suggest the authors to change along the text either the term ‘isotropic’ for ‘homogeneous’ or to call differently the ‘radial’ structures to avoid this semantic problem.
  • Table 1: it is also possible to include a number of effective pulses per spot unit for the linear polarization in the “laser pulse condition”
  • Also for table 1, it is important to include information on the repetition rate, since it has been demonstrated that over certain values, heat accumulation can induce important morphological changes in metallic samples.
  • In line 161, double space before “by”.
  • The irradiation atmosphere of the samples is not indicated. Also, the time after which the irradiated Ti allow samples were used for cell growth characterization. These details go in line with possible contamination by hydrocarbons leading to changes in the wetting properties that might affect cell adhesion.
  • In line 227, “radiation wavelength lambda 1” it is not indicated which is the value (1030 nm?).
  • In line 231, double space before “of”.
  • In figure 1, regarding the depth of the linearly produced LIPSS: is it the average?
  • In line 249 (figure caption): “Mountains Map”.
  • In line 259, double space before “of focal”.
  • In figure 2 caption, it is clear that all the images but the SEM micrographs correspond to the same cells, but it should be indicated that the SEM characterization was done over different cells (or explain why they are different).
  • In line 346, “In many studies,”.

Author Response

(The authors gave the same response as above.)
